# Fourier Transform Infrared Spectroscopy to Measure Cholesterol in Goat Spermatozoa

**DOI:** 10.3390/ani15213107

**Published:** 2025-10-26

**Authors:** N. Cortés-Fernández-de-Arcipreste, A. J. Cardenas-Padilla, A. Alcantar-Rodriguez, A. Vázquez-Durán, A. Méndez-Albores, A. Medrano

**Affiliations:** 1Unidad de Investigación Multidisciplinaria L2 (Reproducción Animal), Departamento de Ciencias Pecuarias, Facultad de Estudios Superiores Cuautitlán, Universidad Nacional Autónoma de México (UNAM), Cuautitlán Izcalli 54714, Mexico; acardenasp@cuautitlan.unam.mx (A.J.C.-P.); alicia.alcantar-reproduccion.animal@cuautitlan.unam.mx (A.A.-R.); amedrano@unam.mx (A.M.); 2Unidad de Investigación Multidisciplinaria L14-A1 (Ciencia y Tecnología de Materiales), Facultad de Estudios Superiores Cuautitlán, Universidad Nacional Autónoma de México (UNAM), Cuautitlán Izcalli 54714, Mexico; almavazquez@comunidad.unam.mx (A.V.-D.); albores@unam.mx (A.M.-A.)

**Keywords:** cryopreservation, ATR-FTIR, goat buck, cholesterol efflux, spectra, spermatozoa

## Abstract

This study standardized the use of attenuated total reflection–Fourier-transform infrared spectroscopy (ATR-FTIR) to measure cholesterol in goat buck sperm. Cholesterol is a component of the sperm plasma membrane that is removed during both physiological (capacitation, to be fertile) and non-physiological (cryopreservation) conditions. For this, three experiments were carried out: (i) determination of the appropriate sperm concentration to detect cholesterol in the FTIR spectrum; (ii) determination of the minimum percentage of viable spermatozoa required to observe at least five spectral bands in common with pure cholesterol; (iii) assessment of cholesterol removal in frozen–thawed spermatozoa. The lowest sperm concentration at which spectral bands were clearly identified was 13 million sperm/mL; regarding viability, the cut-off value was 50%. Five areas of the cholesterol bands decreased in thawed compared to fresh spermatozoa. In conclusion, FTIR is a useful technique to quantify the cholesterol efflux in spermatozoa.

## 1. Introduction

Cryopreservation of spermatozoa is one of the most widely used Assisted Reproductive Techniques (ARTs) in domestic animals. This ART aims to maintain the viability and functionality of spermatozoa after thawing for their subsequent use in artificial insemination [1]. Sperm cryopreservation involves a series of physicochemical phenomena that negatively impact the function and structure of spermatozoa, consequently reducing their fertilizing capacity. The main alterations include the following: (i) induction of “premature capacitation”, similar to the physiological capacitation that occurs naturally in the female reproductive tract, resulting in a reduction in the sperm’s fertile lifespan [2]; (ii) intra-and extracellular formation of ice crystals and the consequent cell dehydration due to water outflow to stabilize solute concentration [3,4]; (iii) changes in membrane permeability due to alterations in the protein–lipid complex [5]; (iv) decreased mitochondrial activity, changes in motility, and chromatin damage [5]; (v) phase transition of phospholipids in the membrane bilayer as temperature changes from liquid–crystalline to gel phase and vice versa, with consequent changes in permeability and elasticity [6]; mobilization of cholesterol from the plasma membrane, leading to an increased membrane fluidity [3,7]; and (vii) oxidative stress caused by drastic temperature changes, osmotic stress, and excessive production of reactive oxygen species (ROS) [6].

The plasma membrane is considered the main organelle that has to face the stress of cryopreservation [8]. The sperm plasma membrane is composed of a phospholipid bilayer, within which integral and peripheral proteins are embedded, featuring hydrophilic head groups and hydrophobic fatty acid chains [3]. This membrane is not static; all its components exhibit some degree of movement in such a way that the higher the concentration of phospholipids, the more fluid the membrane becomes. Cholesterol mobilization from the plasma membrane occurs during sperm maturation and capacitation [7] as well as during cryopreservation; thus, cholesterol content influences plasma membrane stability. When the environmental temperature decreases or increases, lipid–lipid and lipid–protein interactions are modified, leading to an increased membrane rigidity and phase transition from a liquid–crystalline to a gel state [8], as well as cholesterol efflux [7]. The molecular structure of cholesterol (C_27_H_46_O) includes 45 C–H bonds (including five CH_3_ or methyl groups), one O–H, one C–O, one C=C, and 29 C–C bonds [9] (Figure 1).

Attenuated total reflection–Fourier transform infrared spectroscopy (ATR-FTIR) is a technique recently used for the characterization and structural analysis of cells and tissues. FTIR detects molecular vibrations as molecules absorb energy from infrared light. An infrared spectrum consists of a representation of the energy emitted by the molecule (transmittance) versus wavenumber (cm^−1^). Vibrational spectroscopy techniques, such as FTIR and Raman spectroscopy, do not require cell labeling and are based on the detection of intrinsic characteristics of molecular groups. The obtained spectra provide information on the presence and conformation of endogenous biomolecules in cells or tissues [10,11]. In this context, FTIR has been used to assess sperm quality [7], while both Raman and FTIR spectroscopies have been applied to quantify oxidative DNA damage in spermatozoa [12]. Gupta et al. [13] conducted a study to observe the vibrational spectra of cholesterol molecules in the spectral region of 4500 to 500 cm^−1^ using a powdered cholesterol sample. This technique has revealed molecular changes, primarily in lipids and proteins, induced by the in vitro capacitation processes [7]. In addition, FTIR has also been used to detect damage in sperm chromatin; IR bands may indicate chromatin structure, allowing for the evaluation of sperm quality for subsequent use [14]. Furthermore, FTIR may be a useful tool to monitor the efflux of cholesterol from the sperm plasma membrane since it is interesting to quantify such efflux in the different seasons of the year and across individual males, as sperm freezability seems to depend on these factors.

This technique has been applied to sperm from different species. In humans, it has provided insights into the α-helix content, showing that capacitated sperm exhibit increased α-helix signals, whereas a high proportion of β-structures is associated with poor sperm quality [7]. In stallions, heat-induced protein denaturation has been shown to increase extended β-sheet structures and simultaneously decrease α-helix content. Additionally, alterations in the characteristic vibrational bands of the DNA backbone occur as a result of oxidative stress, dehydration, and chromatin condensation. Moreover, the ratio between the asymmetric and symmetric phosphate stretching bands shows a negative correlation with the proportion of spermatozoa exhibiting morphological abnormalities [11,14]. In pigs, different methods have been employed to calculate cholesterol efflux [15]; however, the identification of the representative bands of cholesterol under varying conditions—(i) different sperm concentrations, (ii) varying proportions of live and dead spermatozoa, and (iii) frozen–thawed sperm—has not yet been studied. Therefore, the objective of this study was to standardize the use of ATR-FTIR to measure cholesterol efflux in goat buck spermatozoa.

## 2. Materials and Methods

### 2.1. Animals and Housing

Experiments were conducted at the Multidisciplinary Research Unit, Animal Reproduction Laboratory (L2), Faculty of Higher Studies Cuautitlan, UNAM (19°69’ N). Three Saanen goat bucks (3–4 years old) were used as semen donors during the non-reproductive season (May–August 2024). Animals were housed in the postgraduate pens and their diet consisted of concentrate feed, alfalfa, mineral blocks, and water (ad libitum).

### 2.2. Semen Collection and Sperm Assessment

Glycerol was from Fermont (Mexico City, Mexico), streptomycin and penicillin were from AgroVet (Tlaquepaque, Mexico); all the other chemicals were obtained from Sigma-Aldrich (Darmstadt, Germany). Semen was collected using an artificial vagina. Ejaculates were macroscopically evaluated for volume, color, consistency, appearance, and the presence of strange particles. Immediately after collection and macroscopic assessment, each ejaculate was diluted (1:1 *v*/*v*) in transport medium (Tris 250 mM, glucose 28 mM, citric acid 104 mM, 0.05% streptomycin, and 500 U/mL penicillin) at 37 °C, and samples were transported to the laboratory in about 60 min. Then, the sperm samples were microscopically assessed as previously described by Cardenas-Padilla et al. [16] for mass motility, progressive motility, morphological abnormalities, viability, sperm concentration, capacitation status, and plasma membrane fluidity as follows:

Briefly, for mass motility assessment, a drop of semen diluted 1:1 in transport medium was observed directly under an optic microscope (Leica Microsystems, DM1000, Wetzlar, Germany) at 10× magnification, and a score between 0 and 3 was assigned (modified from Baril) [17].

Progressive motility was subjectively assessed in sperm diluted 1:1 (*v*/*v*) in physiological saline solution (0.9% NaCl *w*/*v*) and observed under an optic microscope (Leica, DM1000) at 10× and 20× magnification. Results were expressed as the percentage of cells showing forward movement.

Viability and sperm morphology were assessed using Eosin–Nigrosine staining. A total of 200 cells [non-stained (alive) and stained (dead)] were counted to calculate the percentage of viability, and the same number of cells were also considered to evaluate sperm morphology: normal, primary, and secondary abnormalities [18].

Acrosome integrity was assessed via direct examination of spermatozoa using a phase-contrast microscope (Leica DM1000 LED) at 100× magnification. Briefly, 50 µL of diluted semen was added to 50 µL of 0.4% glutaraldehyde followed by thorough mixing. For the microscopic observation, a drop of the resulting mixture was placed onto a glass slide and covered with a coverslip. Multiple optical fields were examined, and a total of 200 cells were counted. Spermatozoa exhibiting any acrosomal abnormality were classified as acrosome-damaged.

For sperm concentration, each ejaculate was centrifuged (Centrificient CRM Globe, Chicago, IL, USA) at 750× *g* for 15 min. The supernatant was removed, and the sperm pellet was resuspended in 1 mL of transport medium. The resuspended samples were then diluted 1:200 (*v*/*v*) in 0.4% formalin solution and counted in a Neubauer chamber under an optical microscope (Leica, DM1000) with 40× magnification.

Capacitation status was assessed using the Chlortetracycline (CTC) assay. Briefly, 100 µL of diluted semen was placed in a microcentrifuge tube and mixed (1:1, *v*/*v*) with 100 µL of CTC solution (130 mM NaCl, 20 mM Tris, 805 mM CTC-HTC, and 5 mM cysteine, pH 7.8) for 30 s. The mixture was then mixed with 22 µL of 0.2% glutaraldehyde solution. A drop of the mixture was placed on a glass slide, overlaid with a drop of DABCO, and covered with a coverslip. CTC fluorescence patterns were classified as follows: F uniform fluorescence over the whole head (non-capacitated, acrosome-intact); B—absence of fluorescence in the post-acrosomal region (capacitated, acrosome-intact); and AR—absence of fluorescence over most of the head except for a band of fluorescence in the equatorial segment (acrosome-reacted). A total of 200 spermatozoa were evaluated across different optical fields using a fluorescence microscope (Leica DMLS) at 100× magnification, and the results are expressed as percentages.

Sperm plasma membrane fluidity was assessed by Merocyanine 540 (MC540) assay. Briefly, 140 µL of diluted semen was placed in a microcentrifuge tube and mixed with 10 µL of the MC540 working solution (495 µL of TALP [Tyrode’s albumin-lactate-pyruvate], 5 µL of MC540 stock solution (5 mL of DMSO, 14 mg of MC540)), mixing them gently for 30 s. The mixture was then fixed with 22 µL of 0.4% glutaraldehyde solution. A drop of the mixture was placed on a glass slide, overlaid with a drop of DABCO, and covered with a coverslip. Merocyanine patterns were classified as opaque (low fluidity) or brilliant (high-fluidity–high-binding cells). At least 200 spermatozoa were assessed across different optical fields using a fluorescence microscope (Leica DMLS) at 100× magnification. Inclusion criteria: only those ejaculates showing 2–3 mass motility and ≥60% progressive motility were included in this study.

### 2.3. FTIR Analysis

A Fourier-transform infrared spectrometer Frontier SP8000 (Perkin Elmer, Waltham, MA, USA) equipped with an in-compartment diamond attenuated total reflection (ATR) accessory (DuraSamp1IR II, Smiths Detection, Warrington, UK) was used. Thirty-two sequential scans in the 4500 to 500 cm^−1^ range were collected with a resolution of 4 cm^−1^. The FTIR spectrometer was calibrated using a mid-infrared (MIR) polystyrene traceable reference material (L1365334, Perkin Elmer, Waltham, MA, USA), and a background spectrum was recorded to eliminate atmospheric interference. The FTIR spectrum of the sperm includes not only the contributions of the concentrated cells but also a large contribution of the transport medium; therefore, a spectrum of pure transport medium was also measured under identical conditions and subtracted from the spectrum of the sperm (subtraction factor = 0.3898) using Spectrum 10.4.2 software.

### 2.4. Experimental Design

The standardization of the FTIR technique was carried out in three stages.

Stage 1. Determination of the appropriate sperm concentration to detect cholesterol in the FTIR spectrum.

The appropriate sperm concentration to detect cholesterol flow was determined as follows: (i) Pure cholesterol powder (Cholesterol Sigma Grade ≥ 99%, CAS-No: 5-88-5), weighing about 10 mg, was used to identify the spectral bands. To obtain a clean spectrum, the spectrum of the sperm transport medium was subtracted from each sample. (ii) Seven different sperm concentrations (from 109 to 1 × 10^6^) were tested, and the spectral bands obtained were compared with those of pure cholesterol. The measured areas listed in the tables are expressed in arbitrary units (A.U.), as automatically generated by the OriginPro 8 software (Microcal Software Inc., Northampton, MA, USA).

Seven samples of sperm were obtained on different days; with one ejaculate obtained from each buck per day. The ejaculates were assessed, pooled, and the sperm concentration was estimated (3620 × 10^6^ sperm/mL). Then, 30 µL was taken and centrifuged at 750× *g* for 15 min. The supernatant was discarded, and pellet concentration was determined. Immediately, aliquots of the pellet were taken and diluted in transport medium to obtain the different sperm concentrations shown in Table 1.

Each sperm dilution was assessed using FTIR as follows: 20 µL per drop was placed on the ATR crystal. The resulting spectra were compared with that of pure cholesterol, and the matching bands were identified. The area of each band was measured to determine the dilution at which each band could be clearly observable.

Stage 2. Determination of the minimum percentage of viable spermatozoa required to observe at least five spectral bands in common with pure cholesterol.

The minimum percentage of viable spermatozoa required to observe at least five spectral bands in common with pure cholesterol was determined. In our study, after thawing, the sperm viability is about 40%; thus, we would like to know whether cholesterol efflux can be appropriately measured. For this, several sperm viability percentages (live/dead) were tested: 100/0, 75/25, 50/50, 25/75, and 0/100. To obtain those proportions, two sperm populations, 100% live and 100% dead, were mixed appropriately. To kill the spermatozoa, two techniques were used as follows: (a) hot water at 70 °C, and (b) liquid nitrogen.

One ejaculate from each of three male goats was collected on the same day, assessed, pooled, and centrifuged; sperm pellet concentration was 4300 × 10^6^/mL. Aliquots of 300 µL were taken in Eppendorf tubes and subjected to either hot water (70 °C) for five minutes or liquid nitrogen for two minutes (twice) to kill the spermatozoa. Viability was assessed to ensure complete inactivation (0%), confirmed by Eosin–Nigrosine testing and direct observation of motility absence under the microscope. Different amounts of live and dead spermatozoa were obtained by mixing two sperm populations (100% live, 100% dead) in the appropriate proportions (live/dead): 100/0, 75/25, 50/50, 25/75, and 0/100. Then, each live/dead mix was analyzed by FTIR, and the areas of the previously identified cholesterol bands (from Stage 1) were measured.

Stage 3. Buck sperm cryopreservation.

Spermatozoa were frozen following a two-step protocol described by Cardenas-Padilla et al. [16]. Briefly, sperm were diluted in an Egg Yolk–Tris freezing medium to reach a final concentration of 200 × 10^6^ sperm/mL and 4% of glycerol; diluted sperm were packaged in 0.25 mL plastic straws and sealed with polyvinyl alcohol. Straws were cooled from room temperature to 5 °C in a refrigerator for about 3.5 h. Once at 5 °C, the straws were exposed to nitrogen vapours 4 cm above the liquid nitrogen level for 15 min. The straws were then immersed in liquid nitrogen and stored in a Dewar at −196 °C until required.

To evaluate the use of FTIR for measuring cholesterol efflux in frozen–thawed spermatozoa, eight straws were thawed by immersion in a 37 °C water bath for 30 s, and the sperm were transferred to dry tubes at the same temperature. Sperm quality was assessed microscopically as described above. Subsequently, 50 µL aliquots of semen were placed in Eppendorf tubes for FTIR analysis.

### 2.5. Statistical Analysis

The variable measured included the areas of each identified cholesterol band across different sperm concentrations, different viability percentages from the two methods to kill the sperm (heat and cold), and the sperm characteristics after freeze-thawing. All the stages, including progressive motility, viability, acrosome integrity, capacitation status, and membrane fluidity, were tested for normality and homogeneity of variance. Potential differences were then analyzed by one-way ANOVA using SPSS version 20.0 (IBM Corp., Chicago, IL, USA).

## 3. Results

### 3.1. Stage 1: Determination of the Appropriate Sperm Concentration to Detect Cholesterol in the FTIR Spectrum

#### Cholesterol Spectrum

The spectrum of pure cholesterol is shown in Figure 2. The most representative bands were identified, labeled with different letters, and used for comparison with the spectra of sperm from different goat bucks at varying concentrations. Band assignments were based on the literature (Table 2).

The minimum sperm concentration employed in this experiment was 13 × 10^6^ sperm/mL, at which five spectral bands were clearly identified. To obtain a clean spectrum, the spectrum of the sperm transport medium (Figure 3) was subtracted from each sample.

Figure 4 shows the spectra of sperm at different concentrations with each characteristic cholesterol band identified. A decrease in the intensity and area of each band was observed as sperm concentration decreased.

Six bands representing cholesterol were clearly identified: (1) Band C (X = 2924, Y = 99.5): O–H and C–H bonds (red line). (2) Band J (X = 1453, Y = 98.39): CH_3_, CH_2_, and CH bonds (green line). (3) Band M (X = 1226, Y = 97.34): CH_2_ and C–O bonds (blue line). (4) Band O (X = 1175, Y = 98.61): C–C, CH, and C–O bonds (pink line). (5) Band R (X = 1051, Y = 97.62): C–O bonds (dark green line). (6) Band V (X = 932, Y = 98.52): CH and C–C bonds (gray line).

As the sperm concentrations decreased, those bands also diminished, both in intensity and area (*p* < 0.05). Therefore, the minimum sperm concentration at which those six bands were clearly identified and their areas accurately estimated was 13 × 10^6^ sperm/mL (Table 3).

Taking the values of each band at the highest sperm concentration as 100%, the corresponding values at lower concentrations were expressed as proportional decreases (Table 4).

Based on these results, the minimum sperm concentration at which six well-defined cholesterol bands were observed in the spectrum was 13 × 10^6^ sperm/mL.

### 3.2. Stage 2: Determination of the Minimum Percentage of Viable Spermatozoa Required to Observe at Least Five Spectral Bands in Common with Pure Cholesterol

Significant differences in the FTIR spectra were observed between the different proportions of live/dead spermatozoa in both methods (heat and cold) of killing the spermatozoa (Figure 5).

The six representative cholesterol bands previously identified in Stage 1 were also observed in Stage 2. The wavenumbers and areas of each band were measured; however, band C was not detected in the 0/100 live/dead spermatozoa group.

For the FTIR spectra of different proportions of live/dead (cold-killed) spermatozoa, a decrease was observed in the areas of all cholesterol bands (Figure 6). The identification of the bands and the differences in their areas according to sperm viability are shown in Table 5.

Finally, differences between the two treatments to kill the sperm (heat and cold) for live/dead spermatozoa are summarized in Table 5; band O showed no differences between each of the live/dead percentages.

### 3.3. Stage 3: Buck Sperm Cryopreservation

The spectra of frozen–thawed spermatozoa showed changes in most cholesterol bands (Table 6) compared to the spectrum of 13 × 10^6^ sperm/mL from Stage 1. Figure 7 shows the representative spectrum of the samples after freeze-thawing.

Analysis of fresh versus frozen–thawed spermatozoa showed a significant difference (*p* < 0.05) in progressive motility, viability, morphological abnormalities, and CTC-F, CTC-B, and CTC-AR patterns (Table 7). The proportional increase changes and recovery rates after sperm cryopreservation for each assessed variable are also shown in Table 7.

## 4. Discussion

### 4.1. Stage 1: Determination of the Appropriate Sperm Concentration to Detect Cholesterol in the FTIR Spectrum

The bands from the spectrum of pure cholesterol are consistent with those reported by Vyas and Joshi [19] using cholesterol nanoparticles; by Gupta et al. [13] for a cholesterol sample; and by Paradkar and Irudayaraj [9] for 95% pure cholesterol. According to Gupta et al. [13], the major bands they identified were 2932, 2866, 1464, 1438, 1055, 1022, and 985 (2930, 2866, 1464, 1436, 1054, 1022, and 986 in our study), which are attributed to CH_2_ and CH_3_ groups. Several other bands also show good agreement with our findings, such as the 1671 cm^−1^ band, corresponding to the 1674 cm^−1^ band reported by Gupta et al. [13], which is assigned to C=C stretching.

This band identification allowed us to recognize the most representative bands, which were then used to analyze the spectra at different sperm concentrations and determine the minimum concentration at which they could be clearly observed. The spectra of the different concentrations of goat buck semen show that the intensity of each band progressively decreases. At concentration 3 (13 × 10^6^ spermatozoa/mL), all bands retained an area greater than 10%. In contrast, at the highest concentration (109 × 10^6^ spermatozoa/mL), the O and V bands decreased to less than 10%. The clear identification of the bands was a key factor in selecting this concentration for further analysis.

Cholesterol in spermatozoa has been identified using various techniques, including BODIPY-cholesterol, the filipin method, and commercial cholesterol kits [20,21]. To the best of our knowledge, no systematic comparison between FTIR and other techniques has been conducted. This is the first study to use FTIR to identify cholesterol in buck spermatozoa by characterizing the functional groups of the cholesterol molecule. In contrast, in our work, CTC and MC540 assays were employed simultaneously with the FTIR method.

### 4.2. Stage 2: Determination of the Minimum Percentage of Viable Spermatozoa Required to Observe at Least Five Spectral Bands in Common with Pure Cholesterol

Based on the obtained spectra, the decrease in representative cholesterol bands helps to identify their variation across different proportions of live/dead spermatozoa, as well as the method used to kill them. High and low temperatures affect the sperm plasma membrane in different manners; that is, high temperature increases fluidity, whereas low temperature decreases fluidity [21]. Consequently, the extent of cholesterol loss is expected to vary depending on whether the sperm is subjected to heat- or cold-shock.

Sperm viability is crucial to ensure that the observed changes in cholesterol bands accurately reflect the proportions of live and dead spermatozoa. Bernecic et al. [15] emphasize the importance of assessing viability to correctly evaluate cholesterol efflux.

Spectra of live/dead heat-killed spermatozoa confirmed that the representative cholesterol bands decrease in both area and intensity as the proportion of dead spermatozoa increases. Additionally, significant differences were observed in the areas of all bands; notably, in the 0/100 live/dead sperm sample, the C band disappeared. The C band corresponds to the CH_2_ and CH_3_ functional groups, which are abundant in the cholesterol molecule. Therefore, a structural modification occurring during temperature-induced stress could explain the disappearance of this spectral band.

For cold-killed spermatozoa, the bands decreased in area with increasing proportions of dead spermatozoa. Unlike heat treatment, no bands completely disappeared under cold treatment; in the case of the O band, its area was barely detectable.

When comparing heat and cold treatments to kill the sperm and disrupt the plasma membrane, most spectral bands showed greater areas following heat- than cold-treatment. However, in the 0/100 live/dead sperm sample, the C, J, M, and R bands showed larger areas under cold treatment. These results suggest that the heat treatment has a greater effect on the viability of O-H, CH, CH_3_, and C-O bonds, indicating that it may be a more reliable method for assessing sperm viability in a sample.

### 4.3. Stage 3: Buck Semen Cryopreservation

Semen cryopreservation negatively affects sperm function and structure. An important effect is the cholesterol mobilization from the plasma membrane, leading to increased membrane fluidity [7]. Our spectral analysis revealed a decrease in the areas of five identified cholesterol bands (C, J, M, R, and V), with the V band disappearing entirely. Interestingly, the O band increased in area after cryopreservation. The O band corresponds to CH functional groups found in several lipids, including sphingomyelin and phospholipids [22], and not exclusively in cholesterol. This increase may be due to the use of a Tris–Egg Yolk diluent, which is rich in lipids. The observed proportional decreases, together with statistical analysis (ANOVA) support the utility of FTIR for measuring cholesterol changes associated with efflux induced by cryocapacitation.

Finally, sperm quality decreased as expected after cryopreservation. The proportional increase in capacitated sperm with intact membrane (CTC-B pattern), acrosome-reacted sperm (CTC-AR pattern), and hyper-fluid membranes supports the cholesterol efflux findings observed via FTIR spectroscopy.

## 5. Conclusions

While each spectral band corresponds to a specific functional group within the cholesterol molecule, this study has focused on the bands that appeared most clearly in the sperm recorded spectra. At this stage, we presume that the areas of the cholesterol bands may overlap with the areas of other phospholipids that may confound the results. However, the variation in the areas of the band induced by cryopreservation may allow us to associate the specific efflux of cholesterol, excluding other lipids of the plasma membrane. Although, at this stage, it is difficult to weigh each of those bands to precisely quantify the efflux of cholesterol. Future experiments assessing the movement of other sperm membrane lipids associated with cholesterol may give insights into this subject.

Fourier-transform infrared spectroscopy (FTIR) is a useful technique for identifying cholesterol in goat buck spermatozoa by quantifying the areas of the most representative bands of pure cholesterol. It is a simple and rapid method that does not require any specialized media or stains to detect cholesterol bands. The minimum sperm concentration for reliable assessment is about 13 × 10^6^ cells/mL, allowing the precise quantification of band areas. As the proportion of live sperm decreased, both the areas and the intensity of the bands declined, indicating that FTIR could be used to assess sperm viability. In this initial study, we did not quantify cholesterol in absolute terms (milligrams) but calculated proportional decreases relative to the initial band areas. Finally, the observed decrease in functional group band areas, along with changes in membrane fluidity—assessed by CTC assay and MC540 staining after cryopreservation—support the use of FTIR as evidence for cholesterol efflux.

## Figures and Tables

**Figure 1 animals-15-03107-f001:**
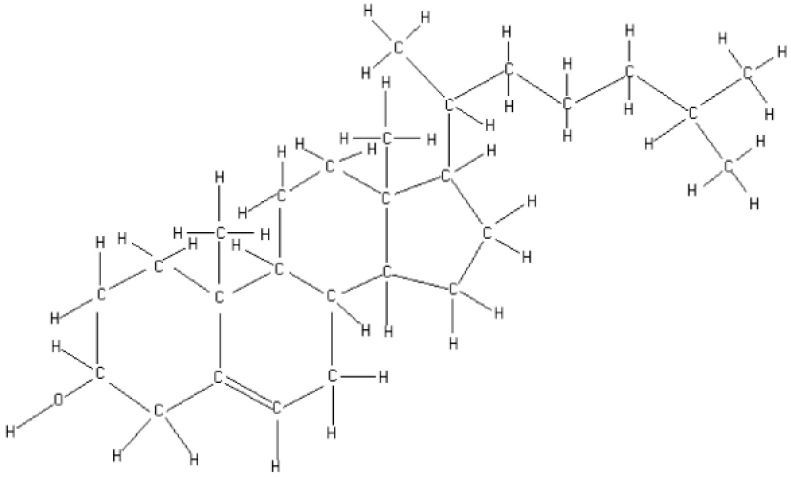
Molecular structure of cholesterol.

**Figure 2 animals-15-03107-f002:**
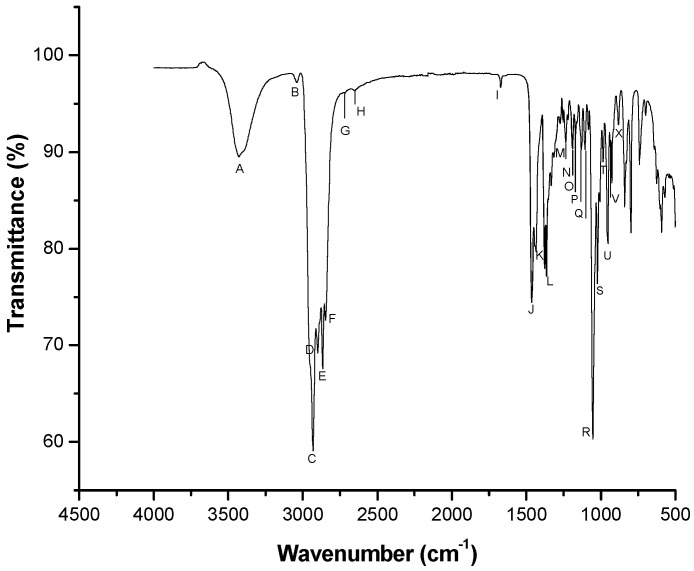
Representative FTIR spectrum of pure cholesterol. Letters indicate the identified spectral bands.

**Figure 3 animals-15-03107-f003:**
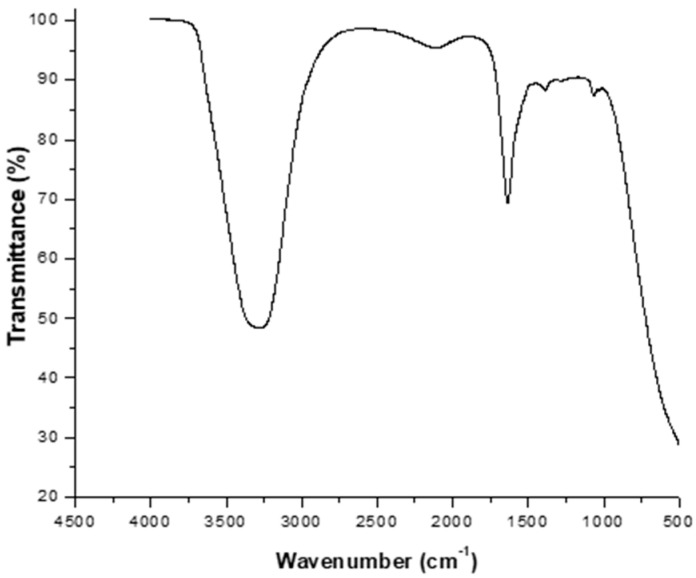
Characteristic spectrum of the sperm transport medium.

**Figure 4 animals-15-03107-f004:**
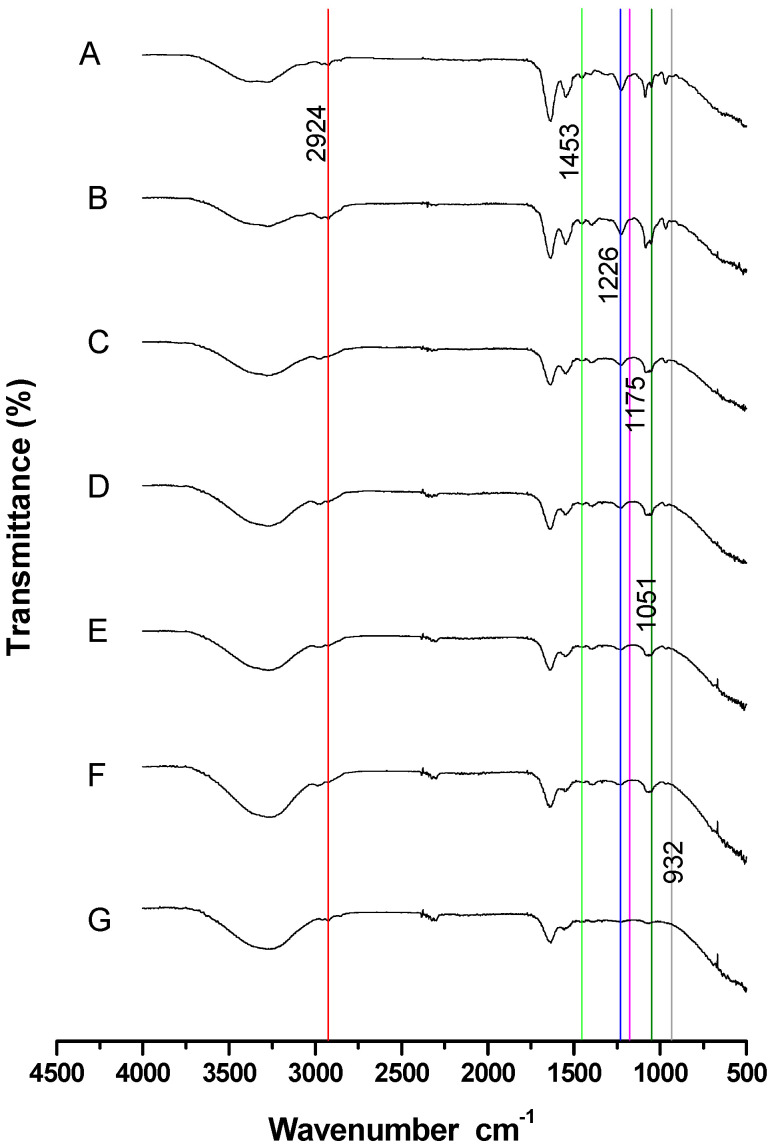
FTIR spectra of goat buck spermatozoa at different concentrations. The top graph represents the spectrum of the initial sperm concentration (sperm/mL): (A) 109 × 10^6^, (B) 54 × 10^6^, (C) 27 × 10^6^, (D) 13 × 10^6^, (E) 6 × 10^6^, (F) 3 × 10^6^, and (G) 1 × 10^6^. Each vertical-colored line corresponds to an identified cholesterol band.

**Figure 5 animals-15-03107-f005:**
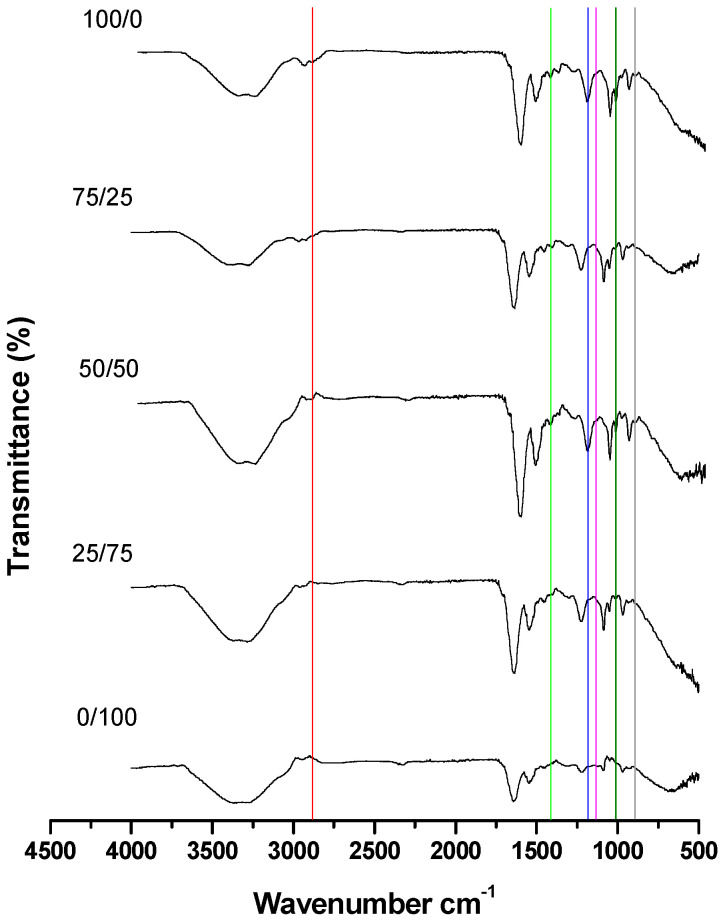
Spectra of different proportions of live/dead (heat-killed) spermatozoa. Each vertical-co-ored line corresponds to an identified cholesterol band.

**Figure 6 animals-15-03107-f006:**
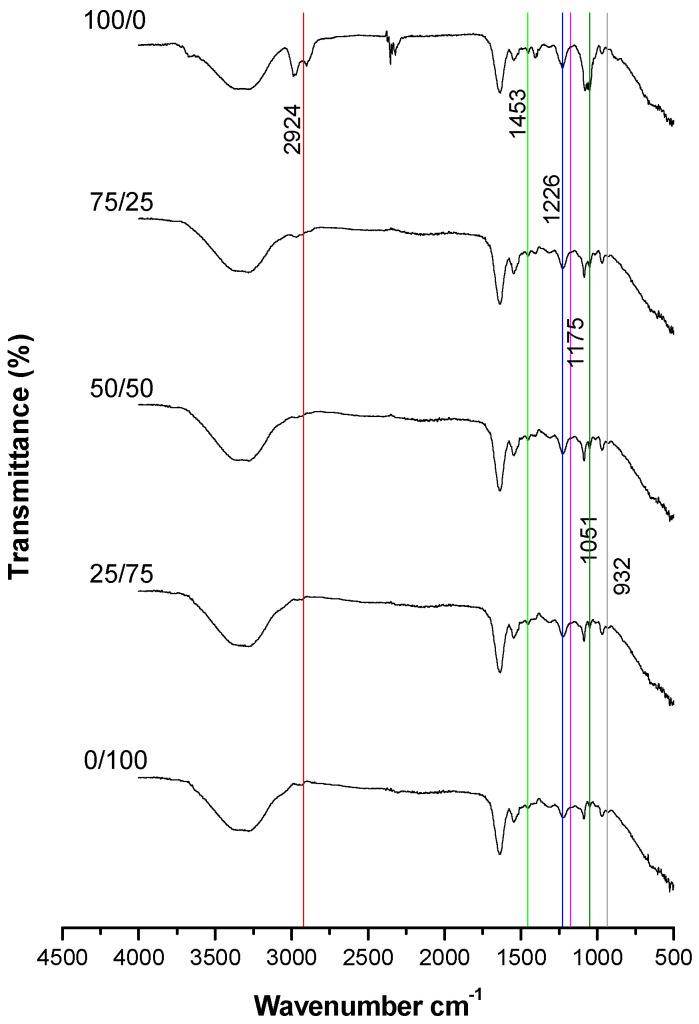
Spectra of different proportions of live/dead (cold-killed) spermatozoa. Each vertical-co-ored line corresponds to an identified cholesterol band.

**Figure 7 animals-15-03107-f007:**
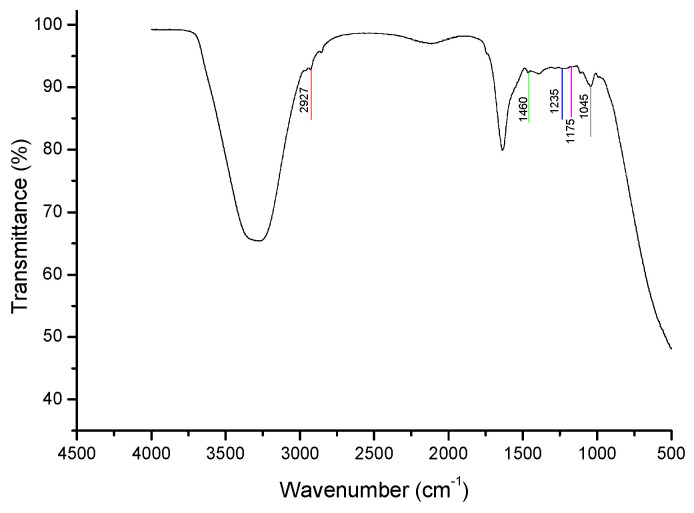
Representative spectrum of frozen–thawed spermatozoa.

**Table 1 animals-15-03107-t001:** Sperm concentrations obtained by diluting the pellet.

Dilution	Concentration (×10^6^ sperm/mL)
1	109
2	54
3	27
4	13
5	6
6	3
7	1

**Table 2 animals-15-03107-t002:** Functional groups corresponding to each cholesterol spectral band, along with their measured areas.

Band	Wavenumber (cm^−1^)	Area (A.U.)	Associated Functional Group
A	X = 3428, Y = 89.52	1545.09	OH
B	X = 3039, Y = 97.17	25.37	CH_2_, CH_3_
C	X = 2930, Y = 59.06	738.78	CH_2_, CH_3_
D	X = 2900, Y = 69.19	42.96	CH_2_
E	X = 2866, Y = 67.58	63.64	CH_2_, CH_3_
F	X = 2847, Y = 72.59	93.31	Nd
G	X = 2718, Y = 96.08	1.53	Nd
H	X = 2653, Y = 96.35	5.82	Nd
I	X = 1671, Y = 96.68	15.08	C=C
J	X = 1464, Y = 74.75	184.61	CH_2_, CH_3_
K	X = 1436, Y = 79.70	81.45	CH_2_, CH_3_
L	X = 1365, Y = 77.17	349.46	CH_2_, CH_3_
M	X = 1235, Y = 90.95	38.21	CH_2_
N	X = 1191, Y = 90.32	53.74	C-C
O	X = 1170, Y = 91.81	11.93	CH
P	X = 1131, Y = 90.24	63.7	CH
Q	X = 1108, Y = 90.26	45.38	CH
R	X = 1054, Y = 60.28	571.54	Deformed ring
S	X = 1022, Y = 7639	72.57	CH
T	X = 986, Y = 89.01	26.32	CH
U	X = 953, Y = 80.50	175.83	CH=CH_2_ Y CH_2_
V	X = 926, Y = 90.59	86.8	CH
X	X = 882, Y = 92.78	45.63	CH

Functional groups found by Gupta et al. [13], Paradkar and Irudayaraj [9], Vyas and Joshi [19]; Nd = not detected. A.U. = arbitrary units.

**Table 3 animals-15-03107-t003:** Comparative analysis of representative cholesterol band areas (A.U.) in goat buck spermatozoa. Values are expressed as mean ± standard deviation.

Band	Sperm Concentration (×10^6^ Cells/mL)							F	*p*
	109	54	27	13	6	3	1		
C	6.28 ± 0.25 a	4.67 ± 0.02 b	4.10 ± 0.11 c	0.99 ± 0.02 d	0.84 ± 0.06 d	0.57 ± 0.06 d	0.56 ± 0.031 d	437.46	0.00
J	9.7 ± 0.1 a	9.1 ± 0.3 a	3.8 ± 0.1 b	3.2 ± 0.07 bc	2.94 ± 0.07 c	2.13 ± 0.07 d	0.76 ± 0.06 e	483.88	0.00
M	61.92 ± 0.2 a	58.3 ± 1.32 b	24.8 ± 0.8 c	17.19 ± 0.4 d	11.75 ± 0.54 e	7.97 ± 0.36 e	3.12 ± 0.13 g	1340.49	0.00
O	0.89 ± 0.05 a	0.51 ± 0.02 b	0.21 ± 0.02 c	0.11 ± 0.02 d	0.05 ± 0.01 d	0.03 ± 0.00 d	0.02 ± 0.00 d	270.30	0.00
R	8.6 ± 0.24 a	6.6 ± 1.20 a	3.44 ± 0.51 b	3.2 ± 0.42 b	2.51 ± 0.05 bc	2.17 ± 0.19 bc	0.11 ± 0.01 c	29.34	0.00
V	18.43 ± 0.30 a	16.04 ± 0.13 b	6.02 ± 0.18 c	3.02 ± 0.09 d	0.21 ± 0.00 e	0.11 ± 0.02 e	0.03 ± 0.00 e	2864.87	0.00

Different letters within the same rows indicate significant differences (*p* < 0.05).

**Table 4 animals-15-03107-t004:** Proportional decrease in representative cholesterol band areas as a function of sperm concentration.

Band	Sperm Concentration (×10^6^ Cells/mL)						
	109	54	27	13	6	3	1
C	100	74.4	65.3	15.8	13.4	9.1	8.9
J	100	93.8	39.2	33.0	30.3	22.0	7.8
M	100	94.2	40.1	27.8	19.0	12.9	5.0
O	100	57.3	23.6	12.4	5.6	3.4	2.2
R	100	76.7	40.0	37.2	29.2	25.2	1.3
V	100	87.0	32.7	16.4	1.1	0.6	0.2

Values of each band in 109 × 10^6^ cells/mL were set as 100%, and values at lower sperm concentrations are expressed as the corresponding proportional decreases.

**Table 5 animals-15-03107-t005:** Comparison of spectral band areas (A.U.) between heat and cold treatments in different proportions of live/dead spermatozoa.

Band	Proportions of Live/Dead Spermatozoa	Heat-Killed Spermatozoa (Mean + Sem)	Cold-Killed Spermatozoa (Mean + Sem)	F	*p*
C	100/0	4.86 ± 0.23 a	0.31 ± 0.02 b	369.98	0.00
	75/25	3.74 ± 0.03 a	0.27 ± 0.05 b	2916.49	0.00
	50/50	1.67 ± 0.32 a	0.19 ± 0.03 b	21.31	0.01
	25/75	0.61 ± 0.16 a	0.15 ± 0.02 a	7.36	0.05
	0/100	0.00 a	0.11 ± 0.01 b	64.47	0.00
J	100/0	11.55 ± 0.72 a	7.22 ± 0.08 b	35.38	0.00
	75/25	9.85 ± 0.71 a	6.43 ± 0.44 b	16.55	0.01
	50/50	9.13 ± 0.03 a	5.21 ± 0.25 b	226.18	0.00
	25/75	7.82 ± 0.12 a	4.90 ± 0.10 b	329.26	0.00
	0/100	0.30 ± 0.08 a	4.85 ± 0.14 b	743.21	0.00
M	100/0	105.33 ± 0.9 a	97.37 ± 0.29 b	69.59	0.00
	75/25	97.54 ± 0.00 a	75.02 ± 0.06 b	12,6124.44	0.00
	50/50	69.42 ± 1.20 a	73.82 ± 0.59 b	10.72	0.03
	25/75	65.68 ± 1.06 a	64.54 ± 0.70 a	0.79	0.42
	0/100	13.93 ± 2.29 a	45.08 ± 2.15 b	157.26	0.00
O	100/0	0.38 ± 0.00 a	0.41 ± 0.03 a	1.29	0.32
	75/25	0.36 ± 0.00 a	0.35 ± 0.07 a	0.03	0.87
	50/50	0.21 ± 0.02 a	0.28 ± 0.04 a	1.66	0.26
	25/75	0.15 ± 0.03 a	0.10 ± 0.01 a	1.80	0.25
	0/100	0.11 ± 0.01 a	0.03 ± 0.00 b	30.25	0.00
R	100/0	18.68 ± 0.97 a	195.76 ± 1.09 b	14,667.64	0.00
	75/25	15.79 ± 0.36 a	12.41 ± 0.34 b	45.51	0.00
	50/50	9.63 ± 0.1 a	10.92 ± 0.49 a	6.47	0.06
	25/75	9.44 ± 0.11 a	9.4 ± 0.14 a	0.04	0.85
	0/100	1.74 ± 0.37 a	6.70 ± 0.39 b	82	0.00
V	100/0	4.64 ± 0.31 a	1.46 ± 0.09 b	96.16	0.00
	75/25	3.77 ± 0.2 a	1.27 ± 0.12 b	110.25	0.00
	50/50	3.69 ± 0.2 a	0.93 ± 0.06 b	169.43	0.00
	25/75	3.66 ± 0.23 a	0.64 ± 0.01 b	172.08	0.00
	0/100	0.43 ± 0.03 a	0.41 ± 0.03 a	0.09	0.77

Mean ± standard deviation. Different letters within rows indicate significant differences (*p* < 0.05).

**Table 6 animals-15-03107-t006:** Comparison of FTIR band areas (A.U.) in fresh and frozen–thawed spermatozoa.

Band	Before Freezing	After Freezing	F	*p*	Proportional Decrease in Cholesterol (%)
C	0.99 ± 0.02	0.67 ± 0.06	3.49	0.07	32.3
J	3.2 ± 0.07 a	0.18 ± 0.03 b	1291.46	0.00	94.4
M	17.19 ± 0.4 a	0.59 ± 0.13 b	2080.98	0.00	96.6
O	0.11 ± 0.02	0.23 ± 0.06	0.46	0.50	209.1 *
R	3.2 ± 0.42 a	1.84 ± 0.2 b	6.57	0.02	42.5
V	3.02 ± 0.09	0			100

Mean ± standard deviation. Different letters within rows indicate significant differences (*p* < 0.05). The final column shows the percentage reduction in cholesterol, considering the 100% reference point before freezing sperm. * Band O was the only one that increased.

**Table 7 animals-15-03107-t007:** Characteristics of fresh and frozen–thawed goat buck spermatozoa.

Variable (%)	Fresh Spermatozoa	Frozen–Thawed Spermatozoa	Recovery Rate
Progressive motility	70.0 ± 7.36 a	36.2 ± 2.39 b	51.7
Viability	69.1 ± 4.18 a	47.4 ± 3.54 b	68.6
Acrosome integrity	78.0 ± 5.12 a	32.06 ± 2.16 b	41.1
CTC-F pattern	72.4 ± 5.74 a	8.2 ± 1.30 b	11.3
			**Proportional Increase**
CTC-B pattern	20.2 ± 6.54	23.0 ± 3.26	113.9
CTC-AR pattern	7.4 ± 2.01 a	68.7 ± 3.37 b	928.4
Hyper-fluid membranes	3.5 ± 0.79 a	55.5 ± 1.94 b	1585.7

Mean ± SEM. Different letters within rows indicate significant differences (*p* < 0.05). Recovery rate = thawed value × 100/fresh value for each variable. Proportional increase = thawed value × 100/fresh value for each variable.

## Data Availability

The original contributions presented in this study are included in the article. Further inquiries can be directed to the corresponding author.

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
