# Peer review of "Fourier Transform Infrared Spectroscopy to Measure Cholesterol in Goat Spermatozoa"

_animals, 2025, doi:10.3390/ani15213107_

Round 1

Reviewer 1 Report

Comments and Suggestions for Authors

I would like to see the commercial brand or trademark of each material used throughout the materials and methods.

Also, Whether in methods or in the graphs, they should include the number of times the experiments were repeated n=x. 

Why does Table 9 show the percentage of thawed sperm with 74% acrosomal membrane integrity but also 68% CTC AR? Aren't these contradictory data? Explain.

In online discussion 423-427, my question is, haven't cholesterol been compared simultaneously in the same sample using FTIR and another of the aforementioned techniques in sperm?
All the bands of the cholesterol spectrum, its radicals and their functions in the sperm should be discussed further or delved into in order to understand the experiment as a whole.

Some tables with numbers could be changed to graphs to follow the results more visually and not so numerical.

Author Response

Paper:  Fourier Transform Infrared Spectroscopy to Measure Cholesterol in Goat Spermatozoa

We want to express or deepest gratitude to the reviewers and editor for their feedbacks. Thanks for your kind observations, we have been able to improve our manuscript greatly, as well as clarify ideas that needed major an improvement in the way they were explained to the reader.

Below the answer to comments (blue font).

Reviewer 1

I would like to see the commercial brand or trademark of each material used throughout the materials and methods.

R= we appreciated the feedback. Glycerol was from Fermont (Mexico), Streptomycin and penicillin were from AgroVet (Mexico), all the others reactive were obtained from Sigma-Aldrich (Germany). This information has been included in the MS, line 124-125.

Also, whether in methods or in the graphs, they should include the number of times the experiments were repeated n=x. 

R= such information has been included in line 205

Why does Table 9 show the percentage of thawed sperm with 74% acrosomal membrane integrity but also 68% CTC AR? Aren't these contradictory data? Explain.

R = Now, we have amended the data, the previous one was wrong. New values have been included in the correspondent table, line 417-418.

In online discussion 423-427, my question is, haven't cholesterol been compared simultaneously in the same sample using FTIR and another of the afore mentioned techniques in sperm?
All the bands of the cholesterol spectrum, its radicals and their functions in the sperm should be discussed further or delved into in order to understand the experiment as a whole.

R= to the best of our knowledge, no systematic comparison between FTIR and other techniques has been conducted. In contrast, in our work CTC and MC540 assays were employed simultaneously to FTIR method. This paragraph has been included in the MS.

While each spectral band corresponds to a specific functional group within the cholesterol molecule, this study has focused on the bands that appeared most clearly in the sperm recorded spectra. At this stage we presume that the areas of the cholesterol bands may overlap with the areas of other phospholipids that may confound the results. However, the variation on the areas of the band induced by cryopreservation may allow us to associate the specific efflux of cholesterol but no others lipids of the plasma membrane. However, at this stage is difficult to weigh each of those bands to precisely quantify the efflux of cholesterol. Future experiments, assessing the movement of other sperm membrane lipids associated to cholesterol, may give insights on this subject. This paragraph has been included in the Discussion, lines 442-447.

Some tables with numbers could be changed to graphs to follow the results more visually and not so numerical.

R= now, we have deleted two tables (5 & 6) because results were already included in Table 7. These numbers correspond to the original submission.

Reviewer 2 Report

Comments and Suggestions for Authors

The manuscript entitled «Fourier Transform Infrared Spectroscopy to Measure Cholesterol in Goat Spermatozoa” i) determination of the appropriate sperm concentration to detect cholesterol in the FTIR Spectrum; ii) determination of the minimum percentage of viable spermatozoa required to observe at least five spectral bands in common with pure cholesterol; iii) assessment of cholesterol removal in frozen-thawed spermatozoa. The results revealed that the lowest sperm concentration at which spectral bands were clearly identified was 13 x 106 sperm/mL. Besides,  viability, the cut off value was 50%: higher values produced spectral bands clearly detect- able whereas in smaller values the band´s áreas decreased sharply making difficult to quantify them. Five areas of the cholesterol bands decreased in thawed compared to fresh spermatozoa; an increase in the proportion of frozen-thawed sperm.

 General comments:

The manuscript is well written even some syntax and editing errors.

There is an excessive number of tables and figures. For instante the tables 5 and 6 were completely combined is tbale 7. It Will be better to remove 5 and 6

Figure captions and table titles are not in the appropriate places.

The Material and methods present some errors

Line 121: for mass motility, the semen should not diluted.

Line 238 : 3.1 Stage I. Determination of the appropriate sperm concentration to detect cholesterol in the FTIR Spectrum: all the texte presented under this title should be placed in the MM section. FTIR of colesterol is known and it should be used as a control.

The authors writ 100/0 live/dead and 100/0 please be consistent and write it in the same way

Reduse the number of tables and figures

This manuscipt could not be publish in its form, it should be carefully reviewed

Author Response

Paper:  Fourier Transform Infrared Spectroscopy to Measure Cholesterol in Goat Spermatozoa

We want to express or deepest gratitude to the reviewers and editor for their feedbacks. Thanks for your kind observations, we have been able to improve our manuscript greatly, as well as clarify ideas that needed major an improvement in the way they were explained to the reader.

Below the answer to comments (blue font).

The manuscript is well written even some syntax and editing errors.

There is an excessive number of tables and figures. For instante the tables 5 and 6 were completely combined is tbale 7. It Will be better to remove 5 and 6

R= we appreciated the feedback. Now, we have deleted two tables (5 & 6) because results were already included in Table 7. These numbers corresponds to the original submission.

Figure captions and table titles are not in the appropriate places.

R= these issues have already been modified.

The Material and methods present some errors

R= now, we have amended those errors.

Line 121: for mass motility, the semen should not diluted.

R= while your statement is accurate, the transport of semen from the collection site to the laboratory necessitates the use of a transport medium. This medium plays a critical role in preserving sperm viability, mitigating thermal shock and providing an essential energy source to sustain cellular activity. We use the assessment of mass motility as a monitor to check the sperm arrive viable to the laboratory; we do not employ those values to make any statistical comparison.

Line 238 : 3.1 Stage I. Determination of the appropriate sperm concentration to detect cholesterol in the FTIR Spectrum: all the texte presented under this title should be placed in the MM section. FTIR of colesterol is known and it should be used as a control.

R= figure 1 has been moved to Introduction (line 84); however, outputs of FTIR related to this experiment have been maintained in Results section.

The authors writ 100/0 live/dead and 100/0 please be consistent and write it in the same way

R= now, we have amended those errors.

Reduse the number of tables and figures

R= now, we have deleted two tables (5 & 6) because results were already included in Table 7. These numbers corresponds to the original submission.

This manuscipt could not be publish in its form, it should be carefully reviewed

Reviewer 3 Report

Comments and Suggestions for Authors

Animals 3859896: Fourier Transform Infrared Spectroscopy to Measure Cholesterol in Goat Spermatozoa.

Fourier Transform Infrared Spectroscopy (FTIR) is a powerful, non-destructive technique for analysing the biochemical composition of cells, including sperm. By detecting vibrational modes of molecular bonds, it generates a unique spectral fingerprint of the sample. In sperm, FTIR enables the assessment of lipid composition and membrane structure, including cholesterol content, which is crucial for membrane fluidity, capacitation, and overall fertility potential. FTIR can be used in capacitation, cryopreservation applications and fertility diagnosing studies.

To begin with, I would like to acknowledge the sound quality of language employed throughout the manuscript. The whole manuscript is constructed with clarity and logical coherence. It is rather easy to read and follow. Nonetheless, there is some inconsistency in the level of detail provided across different sections. Given that this is an original article, it would be advisable to expand on specific areas in greater detail.

SIMPLE SUMMARY

In my opinion, this is one of the weaker sections of the manuscript. I understand the need to keep the summary brief, but this overview does not adequately introduce the topic. Perhaps you could simply list the objectives of the work concisely? Nevertheless, I think it should be rewritten. When mentioning the treatments used, I believe that heat/stress is insufficient.

ABSTRACT

From my point of view, it is much better than the summary. It is appropriately worded and encourages the reader to explore the entire article, skilfully introducing the issues addressed within its content.

In the abstract and the rest of the manuscript, the concentration unit should be corrected, where the number 10 should be raised to the power of 6.

INTRODUCTION

The introduction describes the reason for the study and brings the subject matter closer to the reader. After reading the introduction, I see that the aim of the work seems reasonably constructed. Still, I would expand the introduction to provide a more detailed explanation of Fourier Transform Infrared Spectroscopy and its application in examining spermatozoa of various species (if any). There is a statement: “This technique has been used in sperm from different species; however…” – please provide some information about it. In what way? For what purpose? (Etc.) It would offer a solid foundation for the subject under discussion. I would shorten the paragraph concerning the cryopreservation process and combine it with the paragraph on the specifications of the sperm plasma membrane. I would also remove the sections concerning Raman microspectroscopy, as RM and FTIR spectroscopy both probe molecular vibrations but differ in their sensitivity, sample preparation requirements, and applicability to specific sample types. This article deals strictly with the specifics of the operation and possibilities of implementing the FTIR technique.

MATERIAL AND METHODS

In the sections concerning animals and their housing, as well as semen collection and sperm assessment, some questions came to mind:

Why did you gather the semen in the non-reproductive season? As far as I can recall, the fatty acid composition of the sperm membrane varies with the reproductive season, being more favourable in spring and early summer. During this time, elevated PUFA levels are observed, which are linked to greater cell viability and improved tolerance to cryopreservation.

Why were ejaculates from different individuals pooled? Why not analyse the spectra obtained from spermatozoa derived from individual ejaculates?

What were the volumes of the pooled ejaculates obtained? What were the volumes of the samples tested? I think that such information would greatly simplify the wider application of this method among goat bucks.

In my opinion, the methods used to determine sperm motility, morphology, and viability were somewhat outdated (there are more accurate and faster methods for estimating these indicators), but they are still used because of their repeatability, speed and ease of use.

In the part concerning FTIR analysis and Experimental Design, the information about the Spectrum software should be provided. I think it would also be useful to write how the spectrometer was calibrated and how the pure cholesterol spectrum test was prepared (how did you dissolve the pure cholesterol powder? – such information should be provided). Further, how many times have you repeated spectra from the same sample?

Table 1 - The table has been rearranged and should be corrected, if it is not the result of a different version of Word.

The spermatozoa were killed using two techniques. These should be described in detail.

Another question - how was sperm viability checked and confirmed to be 0%?

RESULTS

The tables, figures, and text have been rearranged and should be corrected if they are not from a different version of Word.

Tables 3 and 4 and 5 and 8 - Should units not be specified in this context?

Whether the calculations concerning the FTIR spectra of different concentrations of sperm goat buck and the proportional decrease of the areas of bands were taken into account in describing statistical calculations?

In my opinion, the data in Table 7, concerning the comparative analysis of spectral band areas, appear to have been incorrectly compiled or compared. I am uncertain whether a one-way ANOVA analysis is applicable in this case.

DISCUSSION

I think it would be advisable to remove the bibliographical references since a numbering system is used.

Despite some earlier gaps in information, the discussion is divided into logically arranged sections. This part is coherent and easy to read.

CONCLUSIONS

Conclusions are formulated in a comprehensible and intelligible manner.

REFERENCES:

The references section lists only 25 original scientific papers. This suggests that the subject matter of this publication is relatively new and largely unexplored. This also suggests that the authors are informed and up-to-date in the field, as many of the references are from the last two decades.

IN SUMMARY:

In my view, this manuscript exemplifies strong scientific writing. The sections are clearly and appropriately structured, and the overall style is consistent and suitable for a scholarly review. The subject matter is scientifically compelling and remains incompletely understood, which underscores its relevance and timeliness. While the manuscript does present certain limitations that warrant attention, it holds promise for contributing novel insights into assessing sperm viability among goat bucks.

Thus, I recommend major revisions to enhance the manuscript’s clarity, focus, and comprehensiveness.

Author Response

We want to express or deepest gratitude to the reviewers and editor for their feedbacks. Thanks for your kind observations, we have been able to improve our manuscript greatly, as well as clarify ideas that needed major an improvement in the way they were explained to the reader.

Below the answer to comments (blue font).

Reviewer 3

Animals 3859896: Fourier Transform Infrared Spectroscopy to Measure Cholesterol in Goat Spermatozoa.

Fourier Transform Infrared Spectroscopy (FTIR) is a powerful, non-destructive technique for analysing the biochemical composition of cells, including sperm. By detecting vibrational modes of molecular bonds, it generates a unique spectral fingerprint of the sample. In sperm, FTIR enables the assessment of lipid composition and membrane structure, including cholesterol content, which is crucial for membrane fluidity, capacitation, and overall fertility potential. FTIR can be used in capacitation, cryopreservation applications and fertility diagnosing studies.

To begin with, I would like to acknowledge the sound quality of language employed throughout the manuscript. The whole manuscript is constructed with clarity and logical coherence. It is rather easy to read and follow. Nonetheless, there is some inconsistency in the level of detail provided across different sections. Given that this is an original article, it would be advisable to expand on specific areas in greater detail.

SIMPLE SUMMARY

In my opinion, this is one of the weaker sections of the manuscript. I understand the need to keep the summary brief, but this overview does not adequately introduce the topic. Perhaps you could simply list the objectives of the work concisely? Nevertheless, I think it should be rewritten. When mentioning the treatments used, I believe that heat/stress is insufficient.

R= simple summary has been rewritten, lines 15-27.

ABSTRACT

From my point of view, it is much better than the summary. It is appropriately worded and encourages the reader to explore the entire article, skilfully introducing the issues addressed within its content.

In the abstract and the rest of the manuscript, the concentration unit should be corrected, where the number 10 should be raised to the power of 6.

R= now, we have amended those errors.

INTRODUCTION

The introduction describes the reason for the study and brings the subject matter closer to the reader. After reading the introduction, I see that the aim of the work seems reasonably constructed. Still, I would expand the introduction to provide a more detailed explanation of Fourier Transform Infrared Spectroscopy and its application in examining spermatozoa of various species (if any). There is a statement: “This technique has been used in sperm from different species; however…” – please provide some information about it. In what way? For what purpose? (Etc.) It would offer a solid foundation for the subject under discussion. I would shorten the paragraph concerning the cryopreservation process and combine it with the paragraph on the specifications of the sperm plasma membrane. I would also remove the sections concerning Raman microspectroscopy, as RM and FTIR spectroscopy both probe molecular vibrations but differ in their sensitivity, sample preparation requirements, and applicability to specific sample types. This article deals strictly with the specifics of the operation and possibilities of implementing the FTIR technique.

MATERIAL AND METHODS

In the sections concerning animals and their housing, as well as semen collection and sperm assessment, some questions came to mind:

Why did you gather the semen in the non-reproductive season? As far as I can recall, the fatty acid composition of the sperm membrane varies with the reproductive season, being more favourable in spring and early summer. During this time, elevated PUFA levels are observed, which are linked to greater cell viability and improved tolerance to cryopreservation.

R= we fully agree with your comments. This work was made in the non-reproductive season because of the FTIR availability; however, we are still working on the assessment of the reproductive season.

Why were ejaculates from different individuals pooled? Why not analyse the spectra obtained from spermatozoa derived from individual ejaculates?

R = this was the first approach to assess cholesterol movement, now we are working on the inter-male variation.

What were the volumes of the pooled ejaculates obtained? What were the volumes of the samples tested? I think that such information would greatly simplify the wider application of this method among goat bucks.

R = in fresh semen the pool was about 3-4 ml; however, the volume used for FTIR was only 20 µl (line 224), this information was already included in the submitted version.

In my opinion, the methods used to determine sperm motility, morphology, and viability were somewhat outdated (there are more accurate and faster methods for estimating these indicators), but they are still used because of their repeatability, speed and ease of use.

R = we agree with you. There are more accurate and faster methods, however, we do not have access to them in our lab, lines 146-152.           

In the part concerning FTIR analysis and Experimental Design, the information about the Spectrum software should be provided. I think it would also be useful to write how the spectrometer was calibrated and how the pure cholesterol spectrum test was prepared (how did you dissolve the pure cholesterol powder? – such information should be provided). Further, how many times have you repeated spectra from the same sample?

R= we thank the reviewer for this important comment. The manuscript has been updated to include details information regarding the FTIR analysis. Specifically, we have now described the software used to acquire and analyse the spectra and the calibration procedure of the spectrometer. Pure cholesterol (Sigma-Aldrich ≥ 99% purity, power form) was used to obtain reference spectra; the cholesterol was measured directly as a power without dissolution. Additionally, we have clarified the number of replicates for spectra acquisition from each sample. These details can be found in the revised MM (FTIR analysis), lines 182-192.

Table 1 - The table has been rearranged and should be corrected, if it is not the result of a different version of Word.

R = Now, we have amended those errors.

The spermatozoa were killed using two techniques. These should be described in detail.

R = Now, we have described in a more detailed the way each technique was performed (lines 238-242).

Another question - how was sperm viability checked and confirmed to be 0%?

R = The 0% viability has confirmed by eosin-nigrosine stain (100% of the sperm were stained), and through directly observation of motility absence, lines 240-242).

RESULTS

The tables, figures, and text have been rearranged and should be corrected if they are not from a different version of Word.

R = Now, we have amended those errors.

Tables 3 and 4 and 5 and 8 - Should units not be specified in this context?

R = the measured areas included in the tables of your concern are in Arbitrary units given automatically by de software OriginPro 8, USA.

Whether the calculations concerning the FTIR spectra of different concentrations of sperm goat buck and the proportional decrease of the areas of bands were taken into account in describing statistical calculations?

R = we thank the reviewer for this comment. The calculations of the FTIR spectra, including the areas of the bands at different sperm concentrations and the proportional decreases relative to the highest concentration, were indeed incorporated into the statistical analysis. Specifically, the band areas and proportional decreases were analysed using ANOVA, and significant differences between concentrations were determined (P < 0.05).

In my opinion, the data in Table 7, concerning the comparative analysis of spectral band areas, appear to have been incorrectly compiled or compared. I am uncertain whether a one-way ANOVA analysis is applicable in this case.

R = we used a one-way ANOVA analysis because several pair comparisons were performed between treatments regarding the areas of different spectral bands. For instance, the areas of the C band from heat- and cold-killed sperm were compared.

DISCUSSION

I think it would be advisable to remove the bibliographical references since a numbering system is used.

R = Now, we have addressed this right observation.

Despite some earlier gaps in information, the discussion is divided into logically arranged sections. This part is coherent and easy to read.

R = we appreciate your comment.

CONCLUSIONS

Conclusions are formulated in a comprehensible and intelligible manner.

R = we appreciate your comment.

REFERENCES:

The references section lists only 25 original scientific papers. This suggests that the subject matter of this publication is relatively new and largely unexplored. This also suggests that the authors are informed and up-to-date in the field, as many of the references are from the last two decades.

R = we appreciate your comment.

IN SUMMARY:

In my view, this manuscript exemplifies strong scientific writing. The sections are clearly and appropriately structured, and the overall style is consistent and suitable for a scholarly review. The subject matter is scientifically compelling and remains incompletely understood, which underscores its relevance and timeliness. While the manuscript does present certain limitations that warrant attention, it holds promise for contributing novel insights into assessing sperm viability among goat bucks.

Thus, I recommend major revisions to enhance the manuscript’s clarity, focus, and comprehensiveness.

Round 2

Reviewer 3 Report

Comments and Suggestions for Authors

The authors have made substantial revisions and some rewrites throughout the text, significantly improving its clarity, structure, and overall accessibility. The presentation of results is clear, the issues raised are understandable, and the stylistic choices are appropriate and effective.

Importantly, the authors have responded constructively to all previous comments, addressing each point with care and precision. Their thoughtful additions and revisions have contributed to a more cohesive and unified manuscript.

SIMPLE SUMMARY

In my opinion, it is now written more clearly.

INTRODUCTION

It is beneficial that the authors included species for which FTIR has been previously used. Nevertheless, I would add two or three sentences about the scope of conducted research and its application, e.g., in humans (α-helix content increases in capacitated sperm, while high percentages of β-structures correlates with poor-quality spermatozoa), and stallions (the formation of extended β-sheet structures is associated with protein denaturation; an increase in β-sheet structures correlates with the decrease in α-helical structures; characteristic bands from the DNA backbone change arise in response to induced oxidative damage, water removal, and decondensation; asymmetric/symmetric phosphate band ratio negatively correlates with the percentage of morphologically abnormal spermatozoa). In pigs, a different method was employed to calculate the cholesterol efflux. I believe such an addition will familiarise people who do not use this technique, but work with animal semen, with FTIR, while also promoting its broader adoption for various applications. However, this is merely my opinion, and I leave it to you to consider it.

I still believe I would remove the sentences regarding Raman microspectroscopy  (and references concerning it), as this was not the focus of the work. Although they are often used together, they are not the same.

MATERIAL AND METHODS

It is very good that the cholesterol structure has been supplemented, along with the names of the companies from which the chemicals were obtained.

All other comments have been considered.

RESULTS

Perhaps it would be worth adding this to the methodology text? “The measured areas listed in the tables are in Arbitrary units, as automatically generated by the software OriginPro 8 (USA).”

All other comments have been considered.

IN SUMMARY:

Given these improvements, I have no further comments regarding the quality or readiness of the work. After making a few minor corrections to the text, I believe the manuscript is ready for publication.

Author Response

We would like to express our deepest gratitude to the reviewers and editor for their comments. Thanks to their kind observations, we were able to significantly improve our manuscript and clarify ideas that required significant improvement in the way they were explained to the reader.

Below are the responses to the comments (blue front).

Reviewer

SIMPLE SUMMARY

In my opinion, it is now written more clearly.

R = We appreciate your comment.

INTRODUCTION

It is beneficial that the authors included species for which FTIR has been previously used. Nevertheless, I would add two or three sentences about the scope of conducted research and its application, e.g., in humans (α-helix content increases in capacitated sperm, while high percentages of β-structures correlates with poor-quality spermatozoa), and stallions (the formation of extended β-sheet structures is associated with protein denaturation; an increase in β-sheet structures correlates with the decrease in α-helical structures; characteristic bands from the DNA backbone change arise in response to induced oxidative damage, water removal, and decondensation; asymmetric/symmetric phosphate band ratio negatively correlates with the percentage of morphologically abnormal spermatozoa). In pigs, a different method was employed to calculate the cholesterol efflux. I believe such an addition will familiarise people who do not use this technique, but work with animal semen, with FTIR, while also promoting its broader adoption for various applications. However, this is merely my opinion, and I leave it to you to consider it.

I still believe I would remove the sentences regarding Raman microspectroscopy  (and references concerning it), as this was not the focus of the work. Although they are often used together, they are not the same.

R= Now, we have removed some references (lines 96-98) and added more information about the use of FTIR (lines 107-121) in the Introduction.

MATERIAL AND METHODS

It is very good that the cholesterol structure has been supplemented, along with the names of the companies from which the chemicals were obtained.

All other comments have been considered.

R= We appreciate your comment.

RESULTS

Perhaps it would be worth adding this to the methodology text? “The measured areas listed in the tables are in Arbitrary units, as automatically generated by the software OriginPro 8 (USA).”

All other comments have been considered.

R = We have added the description regarding the measurement of the areas in to the methodology (lines 208-210).
